# Identification of SNPs Associated with Stress Response Traits within High Stress and Low Stress Lines of Japanese Quail

**DOI:** 10.3390/genes12030405

**Published:** 2021-03-12

**Authors:** Steven Shumaker, Bhuwan Khatri, Stephanie Shouse, Dongwon Seo, Seong Kang, Wayne Kuenzel, Byungwhi Kong

**Affiliations:** 1Department of Poultry Science, University of Arkansas, Fayetteville, AR 72701, USA; sshumake@uark.edu (S.S.); sashouse@uark.edu (S.S.); swkang@uark.edu (S.K.); wkuenzel@uark.edu (W.K.); 2Oklahoma Medical Research Foundation, Oklahoma City, OK 73104, USA; bhuwan-khatri@omrf.org; 3Department of Animal and Dairy Science, Chungnam National University, Daejeon 34134, Korea; seotuna@cnu.ac.kr

**Keywords:** stress response, SNPs, quail, ingenuity pathway analysis

## Abstract

Mitigation of stress is of great importance in poultry production, as chronic stress can affect the efficiency of production traits. Selective breeding with a focus on stress responses can be used to combat the effects of stress. To better understand the genetic mechanisms driving differences in stress responses of a selectively bred population of Japanese quail, we performed genomic resequencing on 24 birds from High Stress (HS) and Low Stress (LS) lines of Japanese quail using Illumina HiSeq 2 × 150 bp paired end read technology in order to analyze Single Nucleotide Polymorphisms (SNPs) within the genome of each line. SNPs are common mutations that can lead to genotypic and phenotypic variations in animals. Following alignment of the sequencing data to the quail genome, 6,364,907 SNPs were found across both lines of quail. 10,364 of these SNPs occurred in coding regions, from which 2886 unique, non-synonymous SNPs with a SNP% ≥ 0.90 and a read depth ≥ 10 were identified. Using Ingenuity Pathway Analysis, we identified genes affected by SNPs in pathways tied to immune responses, DNA repair, and neurological signaling. Our findings support the idea that the SNPs found within HS and LS lines of quail could direct the observed changes in phenotype.

## 1. Introduction

Phenotypic diversities occur due to genetic variations within and between populations. Many genotypic variations arise as the result of genetic mutations. One of these mutations is known as a Single Nucleotide Polymorphism (SNP). SNPs are a common mutation and can be considered synonymous or non-synonymous [1]. Synonymous SNPs will either manifest within a noncoding region of a chromosome or silently within a coding region. Non-synonymous SNPs occur within coding regions of chromosomes and may be either missense or nonsense mutations. Both alter the amino acid at the point of mutation, though in differing ways. Missense mutations substitute one amino acid for another, while nonsense mutations stop translation entirely. Each type of mutation alters the protein produced [2]. These protein alterations may lead to alterations in phenotype. Due to this simple mechanism, SNPs are employed as biomarkers in studies surrounding population genetics [3].

Mitigation of stress is one of the primary goals of poultry producers. Stress has been shown to suppress the immune system’s function, decrease egg production, and slow body growth in chickens [4,5]. Hormones released during stress can affect the production of cytokines and chemokines by leukocytes, which have proinflammatory functions. Immunosuppression resulting from chronic stress has also been shown to decrease the effectiveness of vaccines, which would increase the occurrence of disease with a flock. Two different approaches are used for mitigating stress in chickens: improving the production environment and genetic selection for healthier stress responses [5].

This study aimed to identify genetic variations that could influence differential stress responses using a whole genome resequencing method. Two different lines of Japanese quail bred for differential stress responses were used. SNPs resulting in amino acid changes (nonsynonymous) were identified and their potential effects on functionalities were determined.

## 2. Materials and Methods

### 2.1. Ethics Statement

This study was conducted following the recommended guidelines for the care and use of laboratory animals for the National Institutes of Health. All procedures for animal care were performed according to the animal use protocols that was reviewed and approved by the University of Arkansas Institutional Animal Care and Use Committee (IACUC Protocol #14012).

### 2.2. Birds and DNA Sequencing

For the purposes of this study, two different lines of Japanese quail—a high stress (HS) line and a low stress (LS) line—have been utilized. These two lines have been selected since the 1980s for divergent blood corticosterone response to restraint-induced stress in order to serve as models for stress responses in poultry [6]. The differences between the two lines include higher sociability, lower fear, higher body weight, increased egg production, and a reduced heterophil/lymphocyte ratio in the LS line compared to the HS line. Additionally, the LS line had decreased stress-induced osteoporosis, accelerated onset of puberty, heightened male sexual activity and efficiency, and a decrease in corticosterone levels of approximately one-third when compared to the HS line [6].

The development of HS and LS lines of Japanese quail up to the 12th generation is documented by Satterlee and Johnson (1988) [7]. Each line was selected for its plasma corticosterone response to immobilization [6,7]. An independent random mating condition has since been used for their maintenance [8,9,10]. These research lines were shipped to the University of Arkansas at generation 44 from Louisiana State University and maintained at the Arkansas Agricultural Experimentation Station, Fayetteville, AR [11].

Bird selection and DNA sequencing was performed in direct accordance with Khatri et al. (2019) [6]. Male HS and LS birds were chosen for this study as they have a more stable physiology as a result of having fewer hormonal changes when compared to females. Blood samples (3 ml) from 20 birds each from HS and LS lines were collected. Genomic DNA was purified from each sample using QiaAmp DNA mini kit (Qiagen, Hilden, Germany) following manufacturer’s method. DNA quality was assessed using NanoDrop 1000 (Thermo Scientific, Waltham, MA, USA) and agarose gel electrophoresis. Twelve samples showing highest quality per line were pooled to represent each line. Library preparation and Illumina sequencing for the pooled DNA samples were performed by the Research Technology Support Facility at Michigan State University (East Lansing, MI, USA) using Illumina HiSeq 2 × 150 bp paired end read technology [6].

### 2.3. Data Quality and Assessment

We used the FastQC program (v0.11.6) (https://www.bioinformatics.babraham.ac.uk/projects/fastqc/; accessed on 5 January 2018) to assess the quality of raw reads obtained after sequencing in form of FASTQ files. After quality assessment, the low quality reads were trimmed out using Trimmomatic tool (v0.32) [12]. The clean reads were then mapped onto the Japanese quail reference genome obtained from NCBI (Bioproject PRJNA292031; https://www.ncbi.nlm.nih.gov/genome/113; accessed on 5 February 2019) [6].

For the reference-based genome alignment, the Seqman NGen genome sequence assembly program of the Lasergene software package (DNAStar, Madison, WI, USA) was used. Assembly parameters were as follows: File format, Binary Alignment Map (BAM); mer Size, 21; mer skip query, 2; minimum match percentage, 93; maximum gap size, 6; minimum aligned length, 35; match score, 10; mismatch penalty, 20; gap penalty, 30; SNP calculation method, diploid Bayesian; minimum SNP percentage, 5; SNP confidence threshold, 10; minimum SNP count, 2; minimum base quality score, 5. After assembly, the SeqMan Pro program of the Lasergene package (DNAStar) was used for further analyses [13,14].

### 2.4. SNP Detection and Analysis

Detection and analysis of SNPs followed the methods of Khatri et al. (2018) [14]. The JMP genomics (SAS Institute Inc., Cary, NC, USA) program was used for filtering unique SNPs between the HS and LS lines of Japanese quail. Single nucleotide polymorphism (SNPs) occurring in both HS and LS lines were removed, leaving behind the unique SNPs for each line. To identify highly fixed and homozygous SNPs, the SNPs were filtered based on SNP percentages (SNP%) of SNP% ≥0.90 (for example, number of SNP = 9 of read depth = 10) were chosen. SNPs that induce non-synonymous changes in CDS (coding DNA sequences; protein coding) regions were chosen and unique SNPs in either HS or LS showing ≥10 read depths were selected as reliable SNPs. To reduce false positives, reliable SNPs chosen by criteria described above were confirmed by double-checking the initial assembly results with alignment view in SeqMan Pro program of Lasergene package (DNAStar) [14].

### 2.5. Bioinformatic Pathway Analysis

Genes containing unique SNPs from the HS and LS lines were entered into Ingenuity Pathway Analysis (version 60467501; Ingenuity Systems; Qiagen, Redwood City, CA, USA) for variant effect analyses for functional interpretation of genes including SNPs. The top 5 canonical pathways were chosen based on *p-*value calculated by the Fisher’s exact test, which was provided by the IPA algorithm.

### 2.6. Sorting Intolerant from Tolerant (SIFT) Prediction

The SIFT prediction for amino acid changes to be deleterious was conducted following the instruction (https://sift.bii.a-star.edu.sg/; accessed on 1 March 2021). Briefly, since SIFT database for Japanese quail (*Coturnix japonica*) is not available, a new database for Japanese quail genome was generated using sequences in FASTA format for genome and GFF file retrieved from NCBI (https://www.ncbi.nlm.nih.gov/genome/?term=txid93934[orgn]; accessed on 1 March 2021) following instruction (https://sift.bii.a-star.edu.sg/sift4g/SIFT4G_codes.html; accessed on 1 March 2021). The newly generated database was, then, annotated with .VCF file containing SNP list causing non-synonymous mutations. The SIFT score ≤ 0.05 is considered as deleterious amino acid substitution according to the instruction (https://sift.bii.a-star.edu.sg/sift4g/AnnotateVariants.html; accessed on 3 March 2021).

## 3. Results

### 3.1. Genome Re-Sequencing and Distribution of SNPs

In accordance with Khatri et al. (2019) [6], we performed whole genome resequencing of pooled DNA samples from 12 birds each from HS and LS lines of the quail and produced ~250 and ~257 million reads of 150 bp respectively. Of those, ~85 and ~84 million reads were mapped to the reference genome (NCBI/*C. japonica*) and their respective depth of coverage reached to ~41× and ~42× for HS and LS (Table 1) [6].

Totally, 10,364 SNPs were uniquely found across the HS and LS line of Japanese quail with SNP% ≥ 0.90 and present within the CDS region. The HS line contained 6551 SNPs, and the LS line contained 3813 SNPs. These SNPs were further filtered to remove synonymous SNPs, leaving only those within the CDS region that would induce changes in amino acid sequences. This brought the number of SNPs to 1831 within the HS line and 1055 within the LS line (Appendix A). This list of SNPs was further filtered to include genes that occurred uniquely within each line. There were 980 and 539 uniquely affected genes (which were used for Ingenuity Pathway Analysis) in the HS and the LS line, respectively. The SIFT prediction tool predicted 707 and 374 amino acid changes to be deleterious substitutions in HS and LS lines, respectively (Appendix A). The 20 SNPs from each line with the highest read depths can be found in Table 2 and Table 3, while the occurrence of each type of impact of the unique SNPs can be found in Table 4. A previous study of SNPs within a chicken autoimmune vitiligo model identified similar mutations in coding regions, causing changes in amino acid sequences [13].

### 3.2. Bioinformatics and Pathway Analyses

IPA was used to analyze connections of genes containing SNPs to canonical pathways and changes in molecular interactions.

Glycoprotein VI (*GP6*) Signaling, Signaling by Rho Family GTPases, Rac Signaling, Amyotrophic Lateral Sclerosis Signaling, and Synaptogenesis Signaling were the top 5 canonical pathways found to be significantly affected within the HS line (*p-*value < 0.01) (Figure 1; Appendix A). In the LS line, the canonical pathways found to be significantly affected (*p-*value < 0.05) include *GP6* Signaling, Activation of Interferon-Regulatory Factor (IRF) by Cytosolic Pattern Recognition Receptors, Th17 Activation, Role of *BRCA2* (*BRCA2*, DNA repair associated) in DNA Damage Response, and IL-15 Production (Figure 2; Appendix A).

The Synaptogenesis signaling and Amyotrophic Lateral Sclerosis signaling pathways found in the HS line are both tied to nervous system development and signaling (Appendix A). Affected genes within the Amyotrophic Lateral Sclerosis signaling pathway are involved in regulation of neuronal damage and degeneration, as well as motor neuron apoptosis, while affected genes within the synaptogenesis signaling pathway are tied primarily to cytoskeletal development mechanisms, such as microtubule stabilization and synaptic spine organization and development [15,16,17,18,19].

The *GP6* Signaling pathway is found to contain SNPs in both lines of quail (Figure 3a,b). Each line shows SNPs within collagen, laminin, phosphoinositide 3-kinase (*PI3K*), and protein kinase C (*PKC*) gene families. In the HS line, Protein kinase C delta (*PKCδ*) is also affected. *PKCδ* is part of a signaling pathway with proinflammatory functions within the nervous system [20]. In the LS line, talin 1 (*TLN1*) and inositol 1,4,5-triphosphate receptor type 1 (*ITPR1*) are affected. *TLN1* expression mediates leukocyte adhesion to platelets [21]. *ITPR1*, also known as *IP3R1*, is a regulator of B cell apoptosis [22]. *GP6* is a member of the immunoglobulin superfamily. It is expressed within platelets and their precursor megakaryocytes, and it serves as the major signaling receptor for collagen, which leads to platelet activation and thrombus formation [23,24]. *GP6* is activated by a few different factors, one of which being low shear stress [25,26,27]. Shear stress is one of many factors that can drive gene expression within endothelial cells of the circulatory system, and low shear stress is often an indication of improper blood flow [28].

The Th17 Activation pathway, which includes genes containing SNPs in the LS line, is an integral part of cellular immune response (Figure 4). In addition to controlling differentiation of Th17 cells, this pathway produces IL-17A, IL-17F, IL-21, and IL-22 cytokines [29,30]. Both IL-17 cytokines have pro-inflammatory functions, are important regulators of neutrophil activity, and help mount antifungal immune responses alongside IL-22 [31,32]. IL-21 induces differentiation of B cells and T cells [33]. Similarly, the activation of IRF pathway is also associated with an immune response. This pathway is primarily involved in a host’s response to viral infection, but its activation is nonetheless a result of a stimulated innate immune response (Appendix A) [34].

The canonical pathways observed within the HS line of quail show significant overlap, most of which is attributed to the presence of phosphatidylinositol-4,5-bisphosphate 3-kinase catalytic subunit delta (*PIK3CD*), phosphatidylinositol-4,5-bisphosphate 3-kinase regulatory subunit 3 (*PIK3R3*), and phosphatidylinositol-4,5-bisphosphate 3-kinase regulatory subunit 6 (*PIK3R6*), which are genes found to be active in all five pathways listed in Figure 1. These three genes code for a family of molecules known as phosphoinositide 3-kinases (*PI3Ks*). *PI3Ks* are part of the immune response. These kinases carry out phosphorylation within leukocytes and regulate the activity of Rho GTPases, which are important controllers of leukocyte motility [35].

Found within the Rac signaling, ALS signaling, signaling by Rho GTPases, and synaptogenesis signaling pathways is the gene p21-activated kinase 1 (*PAK1*), which encodes a p21-activated kinase (*PAK*) protein. *PAK* proteins form the bridge between Rho GTPases and the cytoskeleton, leading to changes in cell motility and growth. This bridge is necessary for successful thrombin-mediated platelet activation [36].

Mitogen-activated protein kinase 8 (*MAPK8*), neutrophil cytosolic factor 1 (*NCF1*), and nuclear factor kappa B subunit 1 (*NFKB1*) genes are found within both the signaling by Rho GTPases and Rac signaling pathways. *MAPK8*, also known as c-Jun N-terminal protein kinase (*JNK1*), belongs to the *MAP* kinase family and reacts to cellular stimuli to trigger early gene expression responses (Figure 5). In mice, this specific kinase family has been linked to T cell proliferation, differentiation, and apoptosis [37]. *NCF1* is a subunit of *NADPH* oxidase, an enzyme complex that drives production of superoxide during an immune response [38]. *NFKB1* is part of a protein complex known as nuclear factor kappa B (*NF-κB*) that acts as a transcriptional regulator integral in both innate and adaptive immune response. Issues with *NF-κB* activation can lead to inflammatory diseases [39].

Within the LS line, only one gene, Janus kinase 2 (*JAK2*), is found within multiple pathways. *JAK2* is found within both the Th17 Activation and IL-15 production pathways (Figure 4; Appendix A). *JAK2* codes for a kinase that is part of the *JAK* family of kinases. This kinase is integral for cell responses to interferon γ (IFNγ), a cytokine that is an important part of both innate and adaptive immune responses, thus cementing this gene within the immune response [40].

The LS line of quail contained multiple SNPs at various stages of a DNA damage response that responds to damage caused by ionizing radiation (Figure 6). Both a gene involved in the initial activation complex, Fanconi anemia complementation group F (*FANCF*), and genes involved in each of the end points of the pathway were found to contain SNPs. In the initial activation, *FANCF* was identified. *BRCA2*, Bloom syndrome RecQ-like helicase (*BLM*), SWItch/Sucrose Non-Fermentable (*SWI*/*SNF*), and *E2F* were genes identified at end points of the repair process. This DNA damage response mediates homologous recombination and chromatin remodeling, while also playing a role in tumor suppression [41,42,43].

Diseases and bio functions that were found to be significantly affected within the HS line (*p-*value < 0.05) include cancers, neurological disease, organismal injury and abnormalities, psychological disorders, and developmental disorders. In the LS line, the significantly affected (*p-*value < 0.05) diseases and biofunctions include cancers, and organismal injury and abnormalities, neurological disease, hematological disease, and metabolic disease (Appendix A).

The top physiological systems found to be significantly affected (*p-*value < 0.05) in development and function within the HS line include embryonic development, nervous system development and function, organismal development, tissue development, and cardiovascular system development and function. The top significantly affected (*p-*value < 0.05) systems within the LS line include nervous system development and function, tissue development, embryonic development, organismal development, and organ development (Appendix A).

As the nervous system plays an integral role in stress response signaling, further analysis of its affected developmental functions was performed. The factors involved in nervous system development and functions were listed in Appendix A. The top 10 functions and the related genes affected by SNPs in each line are listed in Appendix A. Significantly affected (*p-*value < 0.01) nervous system developmental functions in the HS line include branching, growth and outgrowth of neurites, neuritogenesis, dendritic growth and branching, proliferation of neuronal cells, polarization of hippocampal neurons, transport of synaptic vesicles, migration of cerebellar granule cells, and development of neurons (Appendix A). Significantly affected (*p-*value < 0.02) nervous system developmental functions in the LS line include function, maturation, and development of neurons, development of central nervous system, myelination of nerves and the sciatic nerve, formation of inhibitory synapse and the brain, long-term potentiation of mossy fibers, and quantity of dendrites (Appendix A).

## 4. Discussion

Our analysis found unique SNPs in pathways tied to a shear stress induced immune response, a DNA damage repair response network, and several other immune response pathways, including cytokine production and regulation and differentiation of immune cells. Additionally, pathways related to nervous system development were found to be significantly affected by SNPs.

Differing mutations within *GP6* Signaling pathway could be an indication that the two lines of quail have different immune responses to changes in blood pressure [44]. It is possible these two lines may experience changes in blood pressure when exposed to stressors, but the LS line could more efficiently minimize the effects that would have affected endothelial cells. The location of SNPs in the *BRCA2* DNA Damage Response is evidence that the LS line may have become better adapted at repairing cellular damage caused by stress. These two factors taken in tandem indicate that the two lines of quail primarily differ in how their bodies have adapted to control the impact of stress at the cellular level. The LS line, when exposed to the same stressors as the HS line, would not show the same negative impacts to productive traits should its physiology sufficiently compensate.

The LS line was found to have SNPs affecting multiple different pathways tied to the immune response. In particular, the presence of SNPs in the Th17 activation pathway and IL-15 production indicate a possibly direct effect on cytokine production. IL-17A, IL-17F, IL-21, and IL-22 are considered signature cytokines within the immune system with various functions each, including regulation and differentiation of immune cells. Changes in regulatory T-cell activity may protect the cells of the LS line of quail, as these cells suppress many harmful effects of immune responses [5].

The HS line also contained SNPs in genes related to immune function across multiple canonical pathways. *GP6* signaling, Rho GTPase signaling, and Rac signaling pathways were found to be directly connected transitively through SNP mutations of *PI3K* genes and Rho GTPases. The connections of these kinases to leukocytes, as well as each pathway’s connection to inflammatory responses, once again show an association between the immune system and the SNP mutations in these birds.

Both lines of quail were selected based on their response to restraint stress, which is a psychological stressor. Psychological stressors are detected by the nervous system, which can, in turn, activate hormonal responses within the endocrine system. The hypothalamo-pituitary-adrenal (HPA) axis is one example of this type of response. The HPA axis is a driving force behind the endocrine response to stress, as it initiates a hormone cascade leading to the release of corticosterone hormone [45]. The HS and LS lines of Japanese quail show divergent blood corticosterone levels, indicating a lasting effect of genetic selection on a significant change in the response of the neuroendocrine system to stress. Our findings show that SNPs are present in genes throughout several nervous system developmental networks, though the extent to which these SNPs could alter nervous system function still remains unclear. Further studies are needed to understand the relationship of these genes containing SNPs to nervous system development associated with the differential stress responses observed in the HS and LS lines of Japanese quail. Comparison of SNPs from the brain transcriptome could also provide more insight into the effect of stress on neuronal gene expression.

## 5. Conclusions

Overall, these findings seem to support the idea that the higher body weight and increased egg production observed in the LS line of Japanese quail may be attributed to differing neuronal development and a stronger protection from stress induced immune responses. When paired with recent findings that have found CNVs affected genes within nervous and endocrine system development, humoral and cell-mediated immune responses, and various metabolisms, our results support the theory that genotypic variation leads to the observed phenotypic diversity between the HS and LS lines of Japanese quail [6]. Additionally, more research directed towards the immune responses and cellular activity of each line of quail could deepen our understanding of the effects chronic stress may have on differently adapted production birds.

## Figures and Tables

**Figure 1 genes-12-00405-f001:**
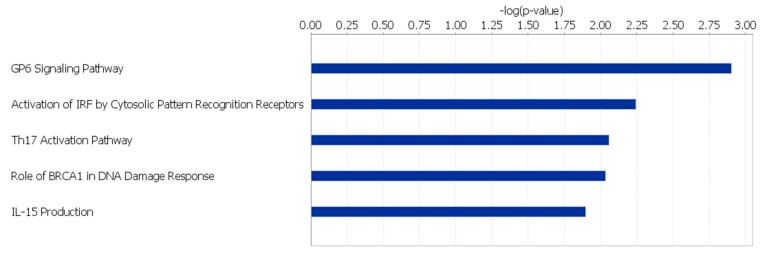
Canonical pathways affected by unique SNPs in HS line of Japanese quail.

**Figure 2 genes-12-00405-f002:**
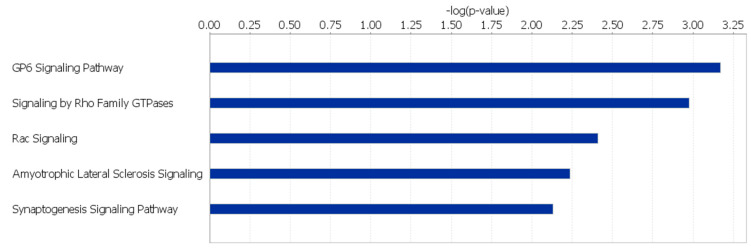
Canonical pathways affected by unique SNPs in LS line of Japanese quail.

**Figure 3 genes-12-00405-f003:**
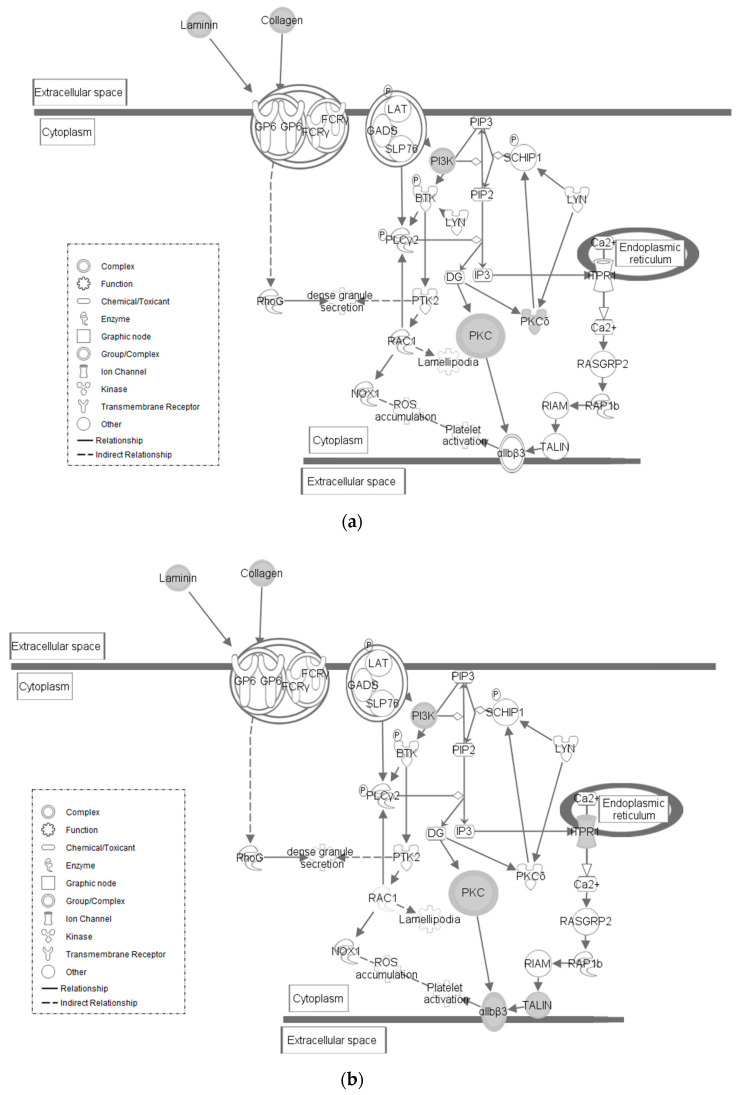
(**a**) *GP6* Signaling Pathway in HS line of Japanese quail. Molecular interactions within the given pathway are shown. Grayed symbols indicate genes containing SNP mutations while white symbols indicate genes that do not contain SNPs but are functionally associated within the pathway. Each symbol represents the given molecule’s function. (**b**) *GP6* Signaling in LS line of Japanese quail. Molecular interactions and symbols within the pathway are the same as described in Figure 3a.

**Figure 4 genes-12-00405-f004:**
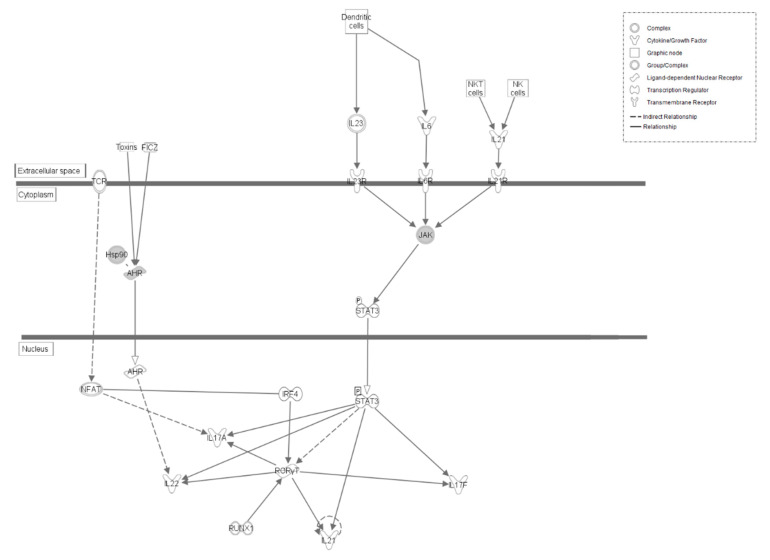
Th17 Activation Pathway in LS line of Japanese quail. Molecular interactions and symbols within the pathway are the same as described in Figure 3.

**Figure 5 genes-12-00405-f005:**
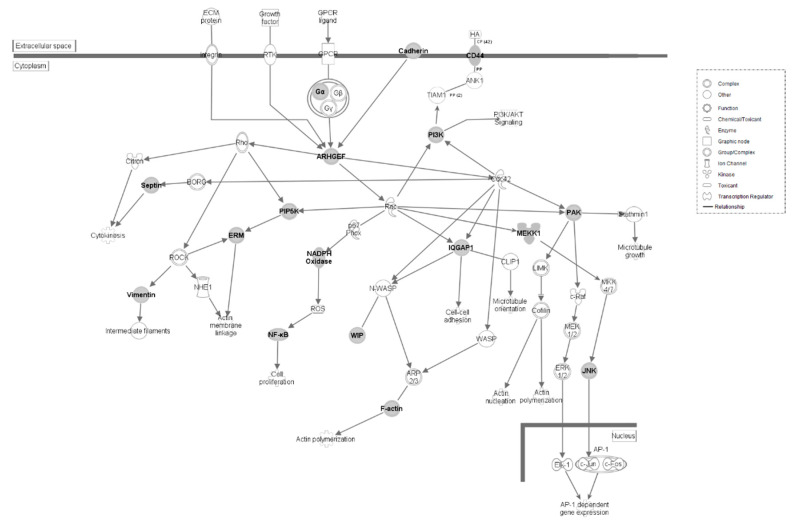
Signaling by Rho Family GTPases and Rac Signaling Pathways in HS line of Japanese quail. Molecular interaction and symbols are the same as described in Figure 3.

**Figure 6 genes-12-00405-f006:**
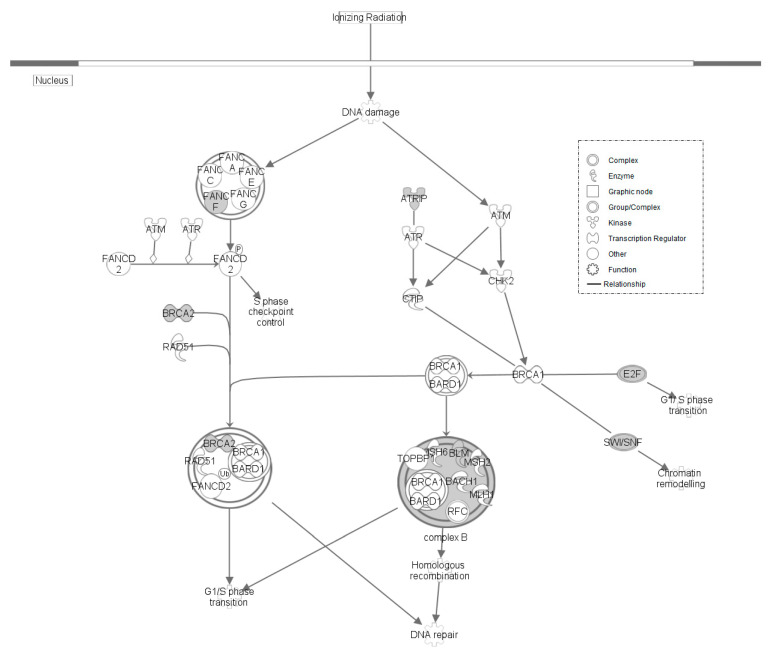
Role of BRCA in DNA Damage Response within LS line of Japanese quail. Molecular interaction and symbols are the same as described in Figure 3.

**Table 1 genes-12-00405-t001:** Sequencing and mapping data of HS and LS lines of Japanese quail [6].

Line	Number of SNPs	Coverage
HS	3,492,469	41.45x
LS	2,872,438	42.59x

**Table 2 genes-12-00405-t002:** Top 20 SNPs with the highest read depths in HS line.

Line	Chr	Ref Pos	Ref Base	Called Base	Impact	SNP %	Feature Name	DNA Change	Amino Acid Change	Depth	A Cnt	C Cnt	G Cnt	T Cnt	Deletion
HS	21	5131174	AC	GT	Non-synonymous	0.99	LOC107323251	c.528_529 > AC	p.W177R	68	-	0	67	0	0
HS	22	2387923	G	A	Non-synonymous	1	R3HCC1	c.1837C > T	p.P613S	56	56	0	-	0	0
HS	8	9560343	-	GCTCAAACAC	Frameshift	1	RNPC3	c.945_946ins	p.N316fs	50	0	0	50	0	0
HS	5	34813334	AT	GC	Non-synonymous	1	TTLL5	c.2643_2644 > GC	p.S882P	50	-	0	50	0	0
HS	8	21795021	A	T	Non-synonymous	1	ZCCHC11	c.2277T > A	p.F759L	49	-	0	0	49	0
HS	12	14314737	A	C	Non-synonymous	0.94	CNTN3	c.966T > G	p.H322Q	48	-	45	0	0	0
HS	1	1.5 × 10^8^	A	G	Non-synonymous	1	LACC1	c.1099A > G	p.T367A	48	-	0	48	0	0
HS	3	91429204	T	A	Non-synonymous	0.94	LOC107312240	c.2044A > T	p.N682Y	47	44	3	0	-	0
HS	7	15772589	A	G	Non-synonymous	1	LOC107316692	c.1156A > G	p.S386G	47	-	0	47	0	0
HS	4	56633029	CC	TT	Non-synonymous	1	PCM1	c.1849_1850 > AA	p.G617N	47	0	-	0	47	0
HS	2	31086012	T	C	Non-synonymous	0.91	PLEKHA8	c.1304T > C	p.L435S	47	0	43	0	-	0
HS	1	70128156	C	G	No-stop	1	EPHB6	c.2447G > C	p..816Sext.?	45	0	-	45	0	0
HS	21	2172350	T	C	Non-synonymous	1	LOC107323499	c.719A > G	p.Q240R	44	0	44	0	-	0
HS	3	60212158	T	C	Non-synonymous	1	REV3L	c.4339T > C	p.S1447P	44	0	44	0	-	0
HS	12	8367256	C	G	Non-synonymous	1	XPC	c.310G > C	p.V104L	44	0	-	44	0	0
HS	7	18690075	T	G	Non-synonymous	0.97	C7H2orf76	c.12A > C	p.L4F	43	0	0	42	-	0
HS	11	9853879	G	A	Non-synonymous	0.93	CCDC79	c.820G > A	p.A274T	43	40	0	-	0	0
HS	3	43357143	A	C	Non-synonymous	1	KATNA1	c.711T > G	p.D237E	43	-	43	0	0	0
HS	2	68425475	A	G	Non-synonymous	1	KIF13A	c.5806T > C	p.W1936R	43	-	0	43	0	0
HS	17	6369984	C	G	Non-synonymous	0.95	INPP5E	c.744C > G	p.F248L	42	0	-	40	0	0

Quail line (High Stress), chromosome (Chr) numbers, reference position (Ref Pos), reference base (Ref Base), called (SNP) base, impact (kinds of protein mutation), SNP%, feature name (gene name), DNA change, amino acid change, Depth, and five columns for SNP counts (Cnt) are indicated.

**Table 3 genes-12-00405-t003:** Top 20 SNPs with the highest read depths in LS line.

Line	Chr	Ref Pos	Ref Base	Called Base	Impact	SNP %	Feature Name	DNA Change	Amino Acid Change	Depth	A Cnt	C Cnt	G Cnt	T Cnt	Deletion
LS	2	32838673	A	G	Non-synonymous	1	PP2D1	c.1616T > C	p.V539A	54	-	0	54	0	0
LS	9	8491817	C	T	Nonsense	1	TRIP12	c.4974G > A	p.W1658.	52	0	-	0	52	0
LS	7	9557668	G	A	Non-synonymous	1	C7H2orf69	c.964G > A	p.V322I	49	49	0	-	0	0
LS	4	2143170	T	C	Non-synonymous	0.91	ACRC	c.242A > G	p.D81G	47	0	43	0	-	0
LS	1	110870115	T	C	Non-synonymous	1	EGFL6	c.401A > G	p.K134R	47	0	47	0	-	0
LS	1	53603348	C	A	Non-synonymous	0.96	PKP2	c.1840G > T	p.V614L	47	45	-	0	0	0
LS	1	64715297	GC	AT	Non-synonymous	0.93	LOC107317569	c.401_402 > AT	p.R134H	45	42	0	-	0	0
LS	6	13674979	A	G	Non-synonymous	0.98	USP54	c.3248A > G	p.E1083G	44	-	0	43	0	0
LS	18	3018090	-	TTG	Inframe insertion	0.93	HEXDC	c.1174insCAA	p.S392del	43	0	0	0	40	3
LS	1	120607492	A	C	Non-synonymous	0.98	LOC107306797	c.374A > C	p.K125T	43	-	42	0	0	0
LS	11	8576282	C	T	Non-synonymous	0.93	TDRD12	c.794C > T	p.S265F	43	0	-	0	40	0
LS	4	30750477	C	G	Non-synonymous	0.95	FRAS1	c.1649G > C	p.R550T	42	0	-	40	0	0
LS	Z	44293804	G	A	Non-synonymous	1	PDZPH1P	c.136G > A	p.A46T	42	42	0	-	0	0
LS	1	19982878	A	T	Non-synonymous	0.98	IQUB	c.1246T > A	p.S416T	41	-	0	0	40	0
LS	1	72163526	A	G	Non-synonymous	1	LOC107319872	c.1789A > G	p.I597V	40	-	0	40	0	0
LS	7	29069125	A	T	Non-synonymous	1	LRP1B	c.6537T > A	p.D2179E	40	-	0	0	40	0
LS	13	4497924	A	G	Non-synonymous	1	RUFY1	c.1657A > G	p.T553A	40	-	0	40	0	0
LS	3	69683334	A	T	Non-synonymous	1	SMIM8	c.29T > A	p.I10N	40	-	0	0	40	0
LS	LGE64	150066	T	C	Non-synonymous	1	LOC107325885	c.47T > C	p.V16A	39	0	39	0	-	0
LS	4	66305765	C	T	Non-synonymous	0.97	RBPJ	c.29G > A	p.R10Q	39	0	-	0	38	0

Quail Line (Low Stress), chromosome (Chr) numbers, reference position (Ref Pos), reference base (Ref Base), called (SNP) base, impact (kinds of protein mutation), SNP%, feature name (gene name), DNA change, amino acid change, Depth, and five columns for SNP counts (Cnt) are indicated.

**Table 4 genes-12-00405-t004:** Occurrence of unique SNP impacts in HS and LS lines of Japanese quail.

Impact	Number of SNPs	Percentage
HS Line:		
Frameshift	46	0.702%
Inframe deletion, conservative	5	0.076%
Inframe deletion, disruptive	5	0.076%
Inframe insertion, conservative	2	0.031%
Inframe insertion, disruptive	4	0.061%
Nonsense	8	0.122%
Non-synonymous	1746	26.652%
No-start	11	0.168%
No-stop	11	0.061%
Synonymous	4720	72.050%
Total	6551	
LS Line:		
Frameshift	33	0.87%
Inframe deletion, conservative	4	0.11%
Inframe deletion, disruptive	3	0.08%
Inframe insertion, conservative	0	0.00%
Inframe insertion, disruptive	4	0.11%
Nonsense	5	0.13%
Non-synonymous	1004	26.56%
No-start	2	0.05%
No-stop	0	0.00%
Synonymous	2758	72.96%
Total	3780	

## Data Availability

All sequence reads described in the manuscript are available under BioProject accession PRJNA706434 in NCBI (https://www.ncbi.nlm.nih.gov/bioproject/PRJNA706434 accessed on 5 March 2021). Illumina sequence reads have been deposited at NCBI’s SRA under the BioProject.

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
