# Peer review of "Identification of SNPs Associated with Stress Response Traits within High Stress and Low Stress Lines of Japanese Quail"

_genes, 2021, doi:10.3390/genes12030405_

Round 1

Reviewer 1 Report

Skumaker et al. conducted genome resequencing for birds from high and low stress lines. The manuscript is well written and provides new insight to the field of animal genetics. Additionally, the analyses seem sound.  

I have only a few minor comments.  

The author identified 2,886, non-synonymous SNPs when they compared two selectively bred populations of Japanese quail. To me, 2886 protein coding changes seems too many. Usually, similar studies found only a few non-synonymous SNPs. However, data analysis on this part seem solid to me. I suggest that a population geneticist and not me should comment on this.  Do other studies find similar number of non-synonymous SNPs? If there are such studies please refer to them.

Line 19. Here mention that you identified around 6 M SNPs in total, 10,364 SNPs on coding regions, of which 2886 were non-synonymous.                    

Line 137: Provide a summary chart for table 2, on “Impact”. What is the proportion of each category? A pie chart?

Line 27:  “Every animal contains the same underlying genetic code within their DNA”. This sentence sounds vague to me. What about genetic variability within and between populations? I would not start the intro with this sentence. This is also the opposite of what you say in next sentence.  

Line 266: In future studies, it would be interesting to combine this data with brain transcriptome of these lines. There you could also test if the synonymous SNPs affect transcription, etc.

Line 277: I suggest to start the discussion with a general statement and a short summary of results, before jumping to details.

Reviewer 2 Report

Recommendation: Minor Revisions

Comments:

Low heritability and difficulty in assessing stress in birds result in low efficiency of selection using the BLUP method. As a consequence, the GBLUP method seems to be hopeful, provided that SNPs with significant influences are selected. Therefore, publications on the basics of genetic determination of stress are very important. The presented paper provides valuable information on genetic variability of quail and the possibility of using SNP in selection for stress responses.

Lines 48-57: The text relates to the description of the research material - please transfer to the part „Materials and Methods”. “2.2. Birds and DNA Sequencing”.

Lines 48 – 60: after transferring information from lines 48-57 to "Materials" part, a short message should be added supporting the research hypothesis given on lines 57-60.

Line 116: which software version was used - please provide reference e.g. IPA software (version 42012434; Ingenuity Systems; Qiagen China Co., Ltd.).

Line 137: please move the table title above the table.

Table 2: please change “Group” to “Line” and add a legend below the table

Table 2 and 3: the legend below the table does not match the abbreviations in the first row of tables - please correct and add HS and LS abbreviations to the legend. In addition, it seems that you can opt out of Contig ID - this information adds nothing.

Table 3: please present the p-value in decimal form

Lines 177-179: I suggest moving the description contained in the last 3 sentences of the table title to the text of the paper. Moreover, the same information is in the box above the graph.

Figure 1a and 1b: both figures show the same except for gray symbols depending on SNPs of different lines. If it is technically possible, please indicate SNP influences depending on the HS and LS lines in one figure.

Reference: some capitalizations of titles of articles are in the list of references, please harmonize towards small fonts when not the first word.

The places for corrections are marked in the manuscript.
